

# Impact of habitat loss on the diversity and structure of ecological networks between oxyurid nematodes and spur-thighed tortoises (*Testudo graeca* L.)

Julieta Benítez-Malvido[1], Andrés Giménez[2], Eva Graciá[2],
Roberto Carlos Rodríguez-Caro[2], Rocío Ruiz De Ybáñez[3],
Héctor Hugo Siliceo-Cantero[1] and Anna Traveset[4]

[1] Laboratorio de Ecología del Hábitat Alterado, Instituto de Investigaciones en Ecosistemas y Sustentabilidad, Universidad Nacional Autónoma de México (UNAM), Morelia, Michoacán, Mexico
[2] Departamento de Biología Aplicada, Facultad de Ciencias Experimentales, Universidad Miguel Hernández, Elche, Spain
[3] Departamento de Sanidad Animal, Facultad de Veterinaria, Campus de Excelencia Internacional Regional "Campus Mare Nostrum", Universidad de Murcia, Murcía, Spain
[4] Global Change Research Group, Institut Mediterrani d'Estudis Avançats (CSIC-UIB), Esporles, Mallorca, Spain

Corresponding author
Julieta Benítez-Malvido,
jbenitez@cieco.unam.mx

## ABSTRACT

Habitat loss and fragmentation are recognized as affecting the nature of biotic interactions, although we still know little about such changes for reptilian herbivores and their hindgut nematodes, in which endosymbiont interactions could range from mutualistic to commensal and parasitic. We investigated the potential cost and benefit of endosymbiont interactions between the spur-thighed tortoise (*Testudo graeca* L.) and adult oxyurid nematodes (Pharyngodonidae order Oxyurida) in scrublands of southern Spain. For this, we assessed the association between richness and abundance of oxyurid species with tortoises' growth rates and body traits (weight and carapace length) across levels of habitat loss (low, intermediate and high). Furthermore, by using an intrapopulation ecological network approach, we evaluated the structure and diversity of tortoise–oxyurid interactions by focusing on oxyurid species infesting individual tortoises with different body traits and growth rates across habitats. Overall, tortoise body traits were not related to oxyurid infestation across habitats. Oxyurid richness and abundance however, showed contrasting relationships with growth rates across levels of habitat loss. At low habitat loss, oxyurid infestation was positively associated with growth rates (suggesting a mutualistic oxyurid–tortoise relationship), but the association became negative at high habitat loss (suggesting a parasitic relationship). Furthermore, no relationship was observed when habitat loss was intermediate (suggesting a commensal relationship). The network analysis showed that the oxyurid community was not randomly assembled but significantly nested, revealing a structured pattern for all levels of habitat loss. The diversity of interactions was lowest at low habitat loss. The intermediate level, however, showed the greatest specialization, which indicates that individuals were infested by fewer oxyurids in this landscape, whereas at high habitat loss individuals were the most generalized hosts. Related to the latter, connectance was greatest at high habitat loss, reflecting a more uniform spread of

interactions among oxyurid species. At an individual level, heavier and larger tortoises tended to show a greater number of oxyurid species interactions. We conclude that there is an association between habitat loss and the tortoise–oxyurid interaction. Although we cannot infer causality in their association, we hypothesize that such oxyurids could have negative, neutral and positive consequences for tortoise growth rates. Ecological network analysis can help in the understanding of the nature of such changes in tortoise–oxyurid interactions by showing how generalized or specialized such interactions are under different environmental conditions and how vulnerable endosymbiont interactions might be to further habitat loss.

## INTRODUCTION

As a consequence of diverse human activities, habitat loss and fragmentation in terrestrial ecosystems have been generally considered as the main threats to the maintenance of biodiversity worldwide (*Brooks et al., 2002*; *Fahrig, 2003*; but see *Fahrig et al., 2019*). This is because native species are susceptible to habitat degradation through local extinctions and the reduction in the size of their populations. One aspect of land-use change that could have an important impact on wild populations and biodiversity is the local extinction of native species caused by changes in the nature of biotic interactions, including both mutualistic (e.g., pollination and seed dispersal) and antagonistic (e.g., pests and pathogens) interactions (*Harvell et al., 2002*; *Patz et al., 2004*; *Aguilar et al., 2006*; *Aguirre & Tabor, 2008*; *Tompkins et al., 2011*; *Trumbore, Brando & Hartmann, 2015*; *Pringle, 2016*). The ecological costs of these changes constitute the loss of species and populations and ecosystem degradation (*Harvell et al., 2002*; *Rizzo et al., 2002*; *Patz et al., 2004*; *Aguilar et al., 2006*; *Aguirre & Tabor, 2008*; *Vurro, Bonciani & Vannacci, 2010*; *Trumbore, Brando & Hartmann, 2015*; *Pringle, 2016*). There are other biotic interactions; however, that have been little explored in the context of habitat loss and fragmentation, such as commensal interactions in which one species benefits whilst the other receives neither benefit nor damage. In some reptiles, for instance, the symbiotic relationships with hindgut oxyurid nematodes (endosymbionts) have been categorized as commensal, mutualistic and/or parasitic (*O'Grady et al., 2005*; *O'Grady & Dearing, 2006*; *Jacobson, 2007*). In fact, for herbivorous reptiles, hindgut oxyurids and their associated gut microbial community (e.g., bacteria) might be parasitic while also playing a positive role in the digestion and assimilation of plant matter, which shows host-parasite-nutrition relationships (*Roca, 1999*; *O'Grady et al., 2005*; *O'Grady & Dearing, 2006*; *Donaldson, Lee & Mazmanian, 2016*; *Midha, Schlosser & Hartmann, 2017*). Some studies have suggested a relationship between the kind of oxyurids and the feeding habits of the reptile host. For instance, the Pharyngodonidae (order Oxyurida) nematodes are endosymbionts typical of reptiles with herbivorous diets (*Roca, 1999*; *O'Grady et al., 2005*; *O'Grady &*

*Dearing, 2006*). Hindgut oxyurids are frequently found in chelonians and they are generally considered to have a commensal or even a mutualistic interaction with their hosts and are rarely pathogenic (*Roca, 1999*; *Mitchell & Diaz-Figueroa, 2005*; *Jacobson, 2007*).

Habitat loss and fragmentation influence local climate, resource availability and also the distribution, abundance and behavior of wildlife species and could therefore change the ecology of parasite and/or endosymbiont transmission (*Kareiva, 1987*; *Patz et al., 2004*; *Aguirre & Tabor, 2008*; *Fuentes-Montemayor et al., 2009*; *Benavides et al., 2012*; *Suzán et al., 2012*; *Benítez-Malvido et al., 2016*; *Pringle, 2016*). Furthermore, there is evidence that habitat loss and fragmentation reduce individual fitness for several animal species (e.g., bats, primates, rodents, toads, lizards and tortoises) through changes in individual body size, as well as decreased growth and reproduction (*Patz et al., 2004*; *Amo, López & Martín, 2007*; *Fuentes-Montemayor et al., 2009*; *Janin, Léna & Joly, 2011*; *Benavides et al., 2012*; *Rodríguez-Caro et al., 2013*). In disturbed habitats, less vigorous and stressed individuals are more vulnerable to parasite infection, whereas an infected individual is more prone to be infected by secondary pathogen species concurrently (*Patz et al., 2004*; *Beldomenico & Begon, 2009*; *Benavides et al., 2012*).

The southeastern Iberian Peninsula, as well as other Mediterranean areas, has been an important speciation hot spot and refuge for the survival of several plant and animal species (*Hewitt, 2011*). Nevertheless, this region has historically been subjected to different human pressures for centuries so that several native species are currently threatened at present because of the increased conversion of natural habitats to agricultural and other land uses (*Sánchez-Zapata & Calvo, 2001*; *Anadón et al., 2006*, *Anadón, Wiegand & Giménez, 2012*; *Anadón et al., 2012*; *Hewitt, 2011*; *Rodríguez-Caro et al., 2017*). In this study, we exclusively used the oxyurids (Pharyngodonidae) present in the hindguts of spur-thighed tortoises, *Testudo graeca* L. (F. Testudinidae) in scrublands of southeastern Spain as a research model to determine whether the level of habitat loss (low, intermediate and high) is associated with the incidence of oxyurids and individual body traits (i.e., carapace length and tortoise weight) and growth rates. The available information has shown that the abundance of *T. graeca* within its original distribution range is strongly associated to environmental factors, such as rainfall and temperature, because as ectotherms, terrestrial tortoises tend to react strongly to variations in their environment (*Kaspari & Valone, 2002*; *Anadón et al., 2006*; *Anadón, Wiegand & Giménez, 2012*; *Anadón et al., 2012*; *Rodríguez-Caro et al., 2016*). Furthermore, such factors have shown strong shifts in fragmented landscapes when compared to continuous natural habitats in different ecosystems elsewhere (*Fuentes-Montemayor et al., 2009*; *Janin, Léna & Joly, 2011*; *Anadón, Wiegand & Giménez, 2012*; *Anadón et al., 2012*; *Laurance et al., 2017*).

Nematodes are the main helminths infecting terrestrial chelonians, and most of them belong to the orders Ascaridida and Oxyurida, which are transmitted by the orofaecal route and sexual contact (*Chávarri et al., 2012*). Ascarid infections are associated with carapace deformities and symptoms of upper respiratory tract disease for captive *T. graeca* individuals (*Rideout et al., 1987*; *Jacobson, 2007*); whereas no ascarid infections have been reported for wild tortoise populations in the study region (*Chávarri et al., 2012*).

In contrast, the role of oxyurids as a limiting factor of tortoise populations is not well understood. In fact, oxyurids can be highly prevalent and are considered to have an almost commensal relationship with their hosts (*Roca, 1999*; *Jacobson, 2007*). In addition, oxyurid infections and prevalence increase with age and/or the tortoise´s size, as has been found for other vertebrates (*Morand & Poulin, 1998*; *Arneberg, 2002*; *Jacobson, 2007*; *Benavides et al., 2012*). For the study region, reductions in growth rates have been reported for *T. graeca* in disturbed habitats as well as differences in nematode communities between wild and captive tortoises (*Chávarri et al., 2012*; *Rodríguez-Caro et al., 2013*). Thus, we expect that tortoises' body traits, growth rates, oxyurid infestation and their interaction are sensitive to habitat loss and fragmentation.

Ecological networks are increasingly being used as a valuable tool for studying the complexity of biotic interactions at a community level, including both mutualistic and antagonistic interactions (*Bascompte et al., 2003*; *Bascompte & Jordano, 2007*; *Benítez-Malvido & Dáttilo, 2015*; *Zarazúa-Carbajal et al., 2016*; *Hernández-Martínez et al., 2019*). Within a given community, different organisms and animal species can interact with each other, which generates complex ecological networks (*Bascompte & Jordano, 2007*). A large number of studies have found no random structures in ecological networks, including nestedness (*Bascompte et al., 2003*; *Bascompte & Jordano, 2007*; *Almeida-Neto et al., 2008*). A network is considered nested if species with fewer interactions (specialists) are connected with species with the most interactions (generalists) in cohesive subgroups (i.e., more generalized networks) (*Guimarães et al., 2006*; *Bascompte & Stouffer, 2009*). Despite the importance and increased knowledge of ecological networks in the literature, most studies have focused on mutualistic interactions (e.g., pollination and seed dispersal; *Bascompte & Jordano, 2013*), whereas antagonistic and commensal interactions involving animals and their pathogens and endosymbionts have received less attention (however, see *Kareiva, 1987*; *Benítez-Malvido & Dáttilo, 2015*; *Zarazúa-Carbajal et al., 2016*; *Hernández-Martínez et al., 2019*).

The general aims of this study were the following: (1) to assess if the level of habitat loss alters the local abundance, species composition, richness and the diversity and structure of ecological networks of oxyurids associated with *T. graeca*, as well as (2) to describe the interaction type between tortoises and oxyurids (i.e., positive, neutral and/or negative) as indicated by the tortoises' body traits and growth rates. We expected that tortoises' body traits, growth rates, oxyurid communities and host-oxyurid interactions to vary with the level of habitat loss, with landscapes of low habitat loss differing from the other landscapes considered (*Patz et al., 2004*; *Qian & Ricklefs, 2006*; *Fuentes-Montemayor et al., 2009*; *Suzán et al., 2012*). It is clear that habitat degradation affects the nature of biotic interactions and ecosystems' function; for herbivorous reptiles, however, there is a limited understanding of the role that hindgut nematodes have on them. Furthermore, conventional community descriptors have not always been successful in distinguishing among habitats with different levels of degradation, which indicates that alterations of the structure and function of ecological communities might be unnoticed in conservation research that does not document and quantify species interactions through

an ecological network approach (*Kareiva, 1987*; *Tylianakis, Tscharntke & Lewis, 2007*; *Kaiser-Bunbury & Blüthgen, 2015*).

## MATERIALS AND METHODS

### Study area

The study region covering the southeastern *T. graeca* Spanish population comprises approximately 2600 km$^2$ of semiarid coastal mountains between the Almeria and Murcia provinces (longitude N38° 23 to N36° 20′ and latitude W0° 30 to W2° 20′; Fig. 1). The climate of the region is semiarid with mild winters, hot summers and low rainfall (200–350 mm/year) during the cooler months of the year. This region has been subjected to human intervention for centuries (*Hewitt, 2011*). Land-use practices are among the most important factors, together with climate, relief and lithology, for determining *T. graeca* presence within its distribution range in southeastern Spain (*Anadón et al., 2006*). Such practices include traditional agricultural crops (dry crops known as secanos), heavily intensified irrigated agricultural lands and highways (*Anadón et al., 2006*). Intensified agricultural practices and fires, as well as habitat reforestation with pines, have been shown to negatively affect *T. graeca* growth, survival and population viability (*Anadón et al., 2006*; *Sanz-Aguilar et al., 2011*; *Rodríguez-Caro et al., 2013*, *2016*, *2017*; *Graciá et al., in press*). Furthermore, in this mosaic of land-use practices, tortoise populations require native scrubland patches larger than ≥1 km$^2$ or 75% scrubland cover at the landscape level to remain viable and prevent local extinctions (*Anadón et al., 2006*). The study sites were located in landscapes with different levels of land use intensification, habitat loss and fragmentation. Nevertheless, traditional agricultural crops (secanos), when not extensive, are recognized as suitable habitats for *T. graeca* populations, and therefore the level of habitat loss was established according to the proportion of suitable habitat for tortoises in the landscape. Each site and/or tortoises' population represents one square kilometer, which correspond to a UTM (Universal Transverse Mercator) cell of 1 × 1 km. In this sense, the landscapes with low habitat loss (sampling sites in Villaltas, Galera and Marinica) comprised on average 95% of suitable habitat, including 84.59% scrubland and 10.75% of traditional agriculture. Landscapes with intermediate habitat loss (Bas Norte, Sierrecica and Chinas) comprised 81.60% of suitable habitat, including 62.21% scrubland and 19.39% traditional agriculture. Finally, landscapes with high habitat loss (Palas, Bas Sur and Misiripalme) encompassed 76.26% of suitable habitat, including 40.71% scrubland and 35.54% of traditional agriculture (Fig. 1). The loss of native scrubland habitat, scrubland fragmentation and land use intensification are related between each other. Native scrubland has been reduced by more than 50% in landscapes with high habitat loss, whereas traditional agricultural fields (secanos) presented a threefold increase therein, when compared to landscapes with low habitat loss. These changes in landscape configuration and composition have been shown to affect resource distribution and availability, challenging the maintenance of tortoises and other animal populations elsewhere (*Anadón et al., 2006*; *Janin, Léna & Joly, 2011*; *Benítez-Malvido et al., 2016*; *Pringle, 2016*; *Rodríguez-Caro et al., 2017*). The probability of *T. graeca* encounter with less than 40% scrubland cover is very low (*Anadón et al., 2006*).

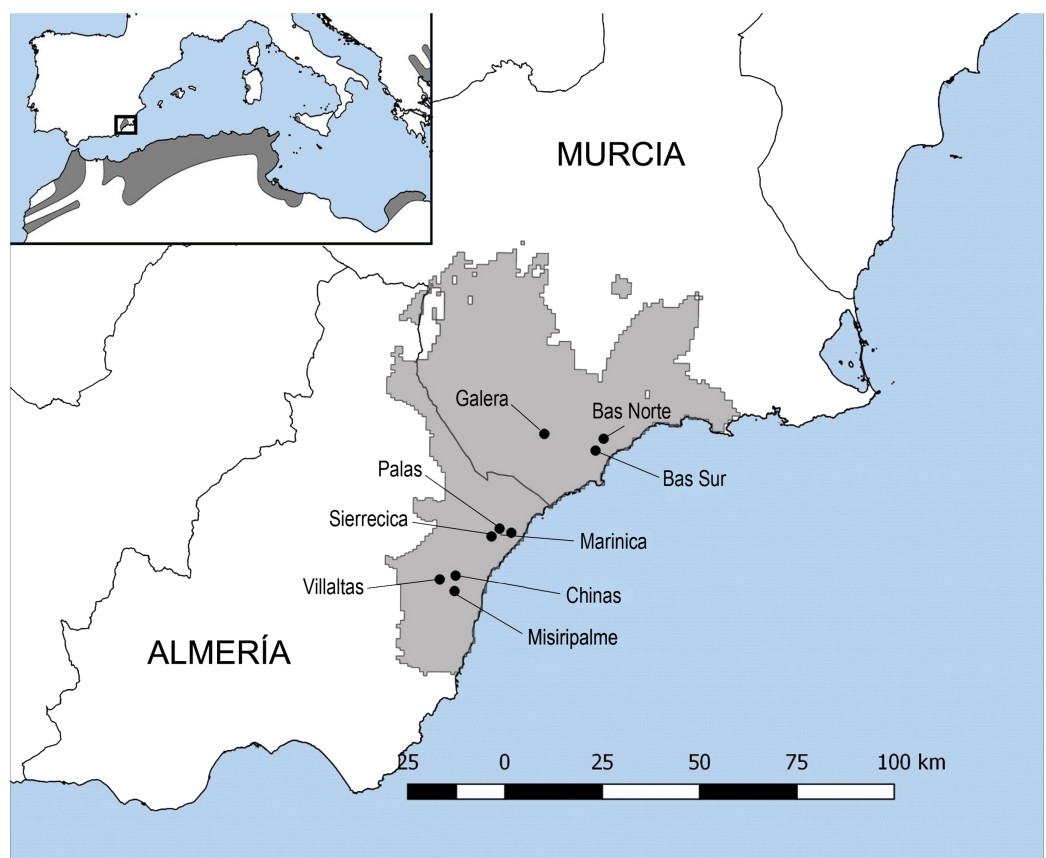

**Figure 1 Map of *Testudo graeca* tortoises sampling sites.** The map shows the distribution of the spur-thighed tortoise (*Testudo graeca*) in the Mediterranean basin. The wild populations sampled for the present study and their localities are indicated with dots: Villaltas, Galera and Marinica, as landscapes with low habitat loss for tortoises; Bas Norte, Sierrecica and Chinas, as landscapes with intermediate habitat loss; and Palas, Bas Sur and Misiripalme, as landscapes with high habitat loss. These wild Spanish populations of *T. graeca* cover approximately 2600 km$^2$ of semiarid coastal mountains between the Almeria and Murcia provinces.                      

## The spur-thighed tortoises and the prevalence of oxyurid nematodes

The Dirección General de Gestión del Medio Natural de la Junta de Andalucía (SGB/FOA/AFR), the Delegación General de Medio Natural de la Comunidad Autónoma de la Región de Murcia (AUT/ET/UND/48/2010) granted the permissions to sample the tortoises and oxyurids. The spur-thighed tortoise (*T. graeca*) is a midsize (up to 46 cm in length, *Werner et al., 2016*) terrestrial tortoise and one of the most widely distributed species of terrestrial tortoises. Although mostly distributed throughout North Africa, the Middle East and part of Eastern Europe, there are some small isolated populations in Western Europe (*Graciá et al., 2017*; Fig. 1). This tortoise species is typically present in arid and semiarid scrublands. The diet of *T. graeca* is based on several parts (i.e., leaves, flowers, fruits and twigs) of wild plants such as alfalfa (*Medicago sativa*), thistles (e.g., *Carthamus* and *Carlina*), dandelion (*Taraxacum*), rosemary (*Rosmarinus officinalis*), etc. (*Andreu, Díaz-Paniagua & Keller, 2000*; *El Mouden et al., 2006*). Furthermore, tortoises may disperse the seeds of the consumed plant species (*Cobo & Andreu, 1988*;

*Andreu, Díaz-Paniagua & Keller, 2000*). The tortoises' diet also includes animal carcasses, slugs and insects, mainly at early life stages. Currently, wild populations of *T. graeca* are severely threatened throughout their whole distribution range because of habitat loss and fragmentation, as well as over-collection, principally in southeastern Spain (*Anadón et al., 2006*, *2012*; *Anadón, Wiegand & Giménez, 2012*; *Pérez et al., 2012*).

From mid March to mid June of 2010, at nine different localities (three for each level of habitat loss), the wild tortoises sighted were captured within standardized transects. Each of the nine sites was visited five times; the sampling time lasted for approximately 2 h 30 min along ca. 1,750 m individual transects. After sampling, specimens were marked and the following individual traits were recorded without harming the tortoises: sex, age estimated from the carapace growth ring (*Rodríguez-Caro et al., 2013*, *2015*), body weight (g) and carapace length (mm). Thereafter, a growth rate score per individual (mm year$^{-1}$, $k$) was calculated using the von Bertalanffy model, the most used to describe chelonians' growth (*Zivkov et al., 2007*; *Macale, Scalici & Venchi, 2009*). This model assumes that the rate of growth declines with age (*Fabens, 1965*) and allows estimating the growth of each individual from its birth to the current measurement time. For further details, see Appendix 1.

For the study of oxyurid communities, individual freshly voided fecal samples were obtained immediately following the handling of the animal or from the cloth bag where animals were placed until they defecated. Thereafter, individuals were released in their capture sites, whereas tortoise fecal samples were maintained at 4 °C until processed within 48 h of collection. Firstly, a saline solution was added to the feces, and under a microscope the adult oxyurids were collected and counted (*McArthur, Meyer & Innis, 2004*; *Traversa et al., 2005*; *Jacobson, 2007*; *Origgi & Paré, 2007*). Secondly, all collected adult oxyurids were washed in distilled water and subsequently stored in 70% ethanol. Thirdly, oxyurids were cleared in Amann lactophenol solution and subsequently identified to the species level following several keys (*Petter, 1966*; *Bouamer & Morand, 2000*, *2002*, *2003*, *2003a*, *2003b*, *2006*; *Bouamer, Morand & Bourgat, 2001a*, *2001b*; *Bouamer, Morand & Kara, 2003*). The number of Oxyuridae eggs and larvae present in the feces were not considered in this study because it was not possible to identify these immature stages to the species level (*Traversa et al., 2005*). Species identification was necessary for the construction of the tortoise-oxyurid ecological networks. Nevertheless, the prevalence of Oxyuridae eggs was the most abundant life stage since 97% of the sampled individuals contained eggs in their feces; whereas oxyurid larvae and adults were present in 36% and 72% of the individuals, respectively. Faecal samples containing adult oxyurids always had oxyurid eggs as well (*Chávarri et al., 2012*). The number of oxyurid eggs, larvae and adults per captured tortoise ranged from 0 to 28,319, from 0 to 1,183 and from 0 to 5,245, respectively. Overall, 14 adult oxyurid species (all within the Oxyurida Family Pharyngodonidae) were isolated in wild populations of *T. graeca*, including the following: *Tachygonetria dentata, T. longicollis, T. macrolaimus, T. conica, T. pusilla, T. numidica, T. robusta, T. setosa, T. palearticus, T. seurati, Alaeuris numidica, Mehdiella stylosa, M. uncinata* and *M. microstoma*.

We sampled a total of 66 wild tortoises from areas with low (26 individuals), intermediate (20 individuals), and high (20 individuals) habitat loss. In landscapes with low habitat loss, we recorded the following: 14 adult females, seven adult males, four subadults and two juveniles. In landscapes with intermediate habitat loss, we recorded three adult females, six adult males, seven subadults and four juveniles. Finally, for the landscapes with high habitat loss, we recorded nine adult females, four adult males, five subadults and two juveniles. Adults were considered as those individuals >8 years old, and subadults were those individuals 5–8 years old. Juveniles were all those tortoises 1–4 years old (*Sanz-Aguilar et al., 2011*).

## Statistical analysis

Because the levels of oxyurid infection and prevalence increase with age (*Chávarri et al., 2012*), we performed the statistical tests, firstly, by pooling all sampled individuals (adults, subadults and juveniles), and secondly, on adult tortoises exclusively. For tortoise density (individuals/ha) we analyzed the entire sampled population. The data were analyzed through generalized linear models (GLMs) by means of the program Statistica 7, and significance was set at the $\alpha = 0.05$ level. Firstly, to test if the level of habitat loss affected tortoise body traits, we used a one-way ANOVA. Body traits values among individual tortoises varied greatly within and across habitat types (e.g., body weight varied between 13 and 1,220 g). Therefore except for growth rates, values were log $(x + 1)$ transformed to improve normality. Because data on oxyurid species richness and abundance were counts (Poisson distribution), we compared the levels of habitat loss by means of log-linear models with a chi-square measure of deviance. When overdispersion was found in the error term of the model, the data were rescaled to make the tests more conservative (*Crawley, 1993*, *2007*). We also tested if oxyurid richness and abundance were related to tortoise density across habitats using log-linear models.

Secondly, to test if tortoise body traits and growth rates were related to oxyurid species richness and abundance across habitat loss categories (the categorical predictor variable with three fixed factors), we compared the linear regression slopes among habitats by using one-way ANCOVAs, with tortoise body traits and growth rates as the dependent variables and oxyurid species richness and abundance as the independent ones (covariates). In total, we performed 12 one-way ANCOVAs, one for each of the two covariates (oxyurid species richness and abundance), by again initially pooling all life stages and then for adults only.

## Oxyurid nematode species dominance

To test whether the level of habitat loss affected oxyurid species relative abundance and/or dominance, we constructed rank/abundance plots for the oxyurid communities present in tortoises at each of the scrubland landscapes (low, intermediate and high habitat loss). For each level of habitat loss, we plotted the relative abundance of each oxyurid species on a logarithmic scale against the species' rank, ordered from the most to the least abundant species (*Magurran, 2004*). To assess whether oxyurid species were evenly distributed (ecological evenness) across the levels of habitat loss, we compared the slopes of the

rank/abundance plots by means of an ANCOVA. The species evenness of a given community is reflected in the slope of the line that fits the rank/abundance plot (i.e., logarithmic series relationship). A steep slope indicates low evenness since the high-ranking species are more abundant than the low-ranking species. A gentle slope indicates higher evenness because species abundances are more similar (*Magurran, 2004*).

## Patterns of species interactions by using a network approach

We used a network approach to investigate the structure of individual-based tortoise–oxyurid networks and their underlying mechanisms. Only individuals with adult oxyurids were considered in the interaction network analysis, mainly adult and subadult tortoises. The number of infected tortoises in the three landscapes were 20, 13 and 15, for low, intermediate and high habitat loss, respectively.

Firstly, we included the presence of different oxyurid species infecting *T. graeca* individuals in each of the landscapes (low, intermediate and high habitat loss) as independent interaction networks. We thus constructed three weighted interaction networks, one for each of the habitat loss conditions. Each individual-based tortoise–oxyurid network was built by an adjacency matrix $A$, where $a_{ij}$ = the number of interactions from an individual tortoise $j$ by the oxyurid species $i$, or otherwise 0 (i.e., the absence of interaction). Thus, in these networks, oxyurid species and tortoises are depicted as nodes, and their interactions are represented by links describing the use of tortoise individuals by oxyurid species. Thereafter, we tested if such networks were nested, a pattern in which oxyurid species with fewer interactions (specialists) are connected with tortoise individuals with the most interactions (generalists) in cohesive subgroups. We used the *WNODF* metric (weighted nestedness metric based on overlap and decreasing fill; *Almeida-Neto et al., 2008*; *Almeida-Neto & Ulrich, 2011*) to estimate the nestedness value of the networks by using ANINHADO software (*Guimarães & Guimarães, 2006*). The *WNODF* metric evaluates whether or not the oxyurids of less selective tortoises represent subsets of the tortoises that are colonized by a broader number of oxyurids (i.e., species and individuals) for each habitat type. This metric ranges from 0 (no nestedness) to 100 (perfect nestedness). It reduces the chance of overestimating the degrees of nestedness in ecological networks and is less prone to Type I statistical error (*Bascompte & Jordano, 2013*). We estimated the significance of nestedness with a Monte Carlo procedure, generating 1,000 random matrices from the original matrix by using the Null Model II (*Bascompte et al., 2003*). In this null model, the probability of occurrence of an interaction is proportional to the number of interactions of both oxyurid species and tortoise individuals (*Bascompte et al., 2003*). Secondly, we considered other network parameters in the analysis for each habitat type as follows (*Kaiser-Bunbury & Blüthgen, 2015*): (i) connectance (the proportion of realized links from the pool of all possible links between the species of a network). Higher connectance indicates greater community stability, reflecting a more uniform spread of interactions among the species in the community; and (ii) interaction diversity (*ID*). This metric is derived from the Shannon diversity index and ranges from 0 (no diversity) to infinity (*Blüthgen, Menzel & Blüthgen, 2006*). Interaction diversity is the quantitative analog to the total number of links. Higher

*ID* implies higher community stability. Finally, we considered (iii) the specialization index or resource selectivity at the network level ($H_2'$). This selectivity index ranges from 0 (extreme generalization) to 1 (extreme specialization) and is highly robust regarding changes in sampling intensity and the number of interacting species (*Blüthgen, Menzel & Blüthgen, 2006*). High specialization indicates the high dependency of each oxyurid species on a few exclusive tortoise partners. Low specialization indicates higher functional redundancy (*Kaiser-Bunbury & Blüthgen, 2015*).

In addition, by using GLMs, we tested if tortoise body traits and growth rates explained network parameters at the individual level for each habitat type (*Kaiser-Bunbury & Blüthgen, 2015*). We used species-degree (the number of oxyurid species infecting each tortoise) and species-specialization (i.e., Blüthgen's *d'*, a measure of the exclusiveness of a species' partner spectrum compared with other species in the network, sensu *Kaiser-Bunbury & Blüthgen, 2015*) as response variables. Species with low species-level specialization indicates opportunistic partner selection compared with other species in the network. For the parameter species-degree, a Poisson type error was used with a log-link function, while for species-specialization, simple linear regression models with Gaussian type errors were used. Network features and plots were obtained by using the Bipartite package in "R" (*Dormann, Fruend & Gruber, 2019*; *R Development Core Team, 2014*).

## RESULTS

Overall, we found that the level of habitat loss affected tortoise–oxyurid interactions with consequences for tortoise growth rates, though not for tortoise weight and carapace length. Moreover, tortoise density was not related to the richness and abundance of oxyurids. Not all sampled tortoises were infested by adult oxyurids across landscapes, with 27%, 35% and 25% of individuals being free from oxyurids, in low, intermediate and high habitat loss, respectively. The number of oxyurid species varied between 0 and 10 species across tortoises.

### Habitat loss, body traits and prevalence of oxyurid nematodes

On the one hand, the one-way ANOVAs and the analysis of deviance showed, through pooling all life stages and by considering only adults, that the level of habitat loss had no significant effect on tortoise body traits and density, nor on oxyurid richness and abundance (Table 1). On the other hand, species richness and the abundance of oxyurids were significantly associated with growth rates depending on the level of habitat loss, which was evaluated both by pooling all life stages and then focusing on adults. Considering all life stages, the ANCOVA tests showed a significant, positive association between oxyurid species richness with tortoise growth rates, but only in landscapes with low habitat loss, as shown by the *habitat loss level x oxyurid species richness* interaction term (richness, $F_{2, 58} = 3.46$, $P = 0.038$) (Fig. 2). Considering adult tortoises only, oxyurid infestation and growth rates were not consistently associated across habitat types (Fig. 2). In landscapes with high habitat loss, tortoises sustained greater growth rates than those with low and intermediate habitat loss (richness, $F_{1, 2} = 6.39$, $P = 0.0042$; abundance,

**Table 1 Tortoise traits, growth rates and oxyurid nematode infestation.** Body traits (mean ± SE), density and oxyurid nematode infestation (i.e., species richness and abundance) of spur-thighed tortoise populations (*Testudo graeca*) at scrubland landscapes with different levels of habitat loss in southern Spain. The values in bold are those exclusively from adult tortoises. There was no significant difference among levels of habitat loss on any variable.

| Body traits and nematode infestation | Low (N = 26) (N = 21) | Intermediate (N = 20) (N = 9) | High (N =20) (N = 13) |
|---|---|---|---|
| Overall tortoise density (individuals/ha) | 2.57 ± 1.19 | 2.26 ± 0.66 | 0.83 ± 0.07 |
| Weight (g) | 431.79 ± 83.04 **527.00 ± 47.90** | 267.36 ± 59.41 **486.00 ± 75.78** | 410.10 ± 91.13 **534.91 ± 59.95** |
| Carapace length (mm) | 129.95 ± 6.86 **131.12 ± 4.01** | 123.44 ± 7.45 **127.10 ± 6.14** | 132.00 ± 7.84 **128.45 ± 5.45** |
| Growth rate (mm year$^{-1}$) | 0.115 ± 0.016 **0.131 ± 0.0149** | 0.085 ± 0.019 **0.123 ± 0.019** | 0.124 ± 0.018 **0.157 ± 0.025** |
| Adult oxyurid species richness | 2.59 ± 0.567 **3.82 ± 0.54** | 2.60 ± 0.66 **5.57 ± 1.31** | 3.55 ± 0.66 **5.27 ± 0.90** |
| Abundance of adult oxyurids | 303.63 ± 184.20 **480.82 ± 308.32** | 200.05 ± 214.01 **561.00 ± 311.05** | 360.30 ± 214.01 **614.27 ± 468.36** |

$F_{1, 2} = 4.42$, $P = 0.019$), whereas the *habitat loss level x oxyurid infestation* interaction term was significant for both species richness ($F_{2, 35} = 6.17$, $P = 0.0051$) and oxyurid abundance ($F_{2, 35} = 4.24$, $P = 0.022$) (Fig. 2). While growth rates were positively associated to oxyurid infestation in landscapes with low habitat loss, the opposite was found in landscapes with high habitat loss; at intermediate habitat loss, no association existed between the two variables (Fig. 2).

## Habitat loss and oxyurid nematode species dominance

The oxyurid community for the three levels of habitat loss followed a log series model of a small number of abundant species and a large proportion of rare species. The adult oxyurids most commonly found in the feces were of the species *T. dentata* and *T. longicollis* (Fig. 3). The following patterns emerged in the rank-abundance plots: (1) all levels of habitat loss showed the same dominant oxyurid species; (2) although rare species changed ranks across the three levels of habitat loss, they were consistently the same species; and (3) the landscapes with high habitat loss presented one exclusive species, *T. seurati*, whereas *T. palearticus* was absent from this landscape type (Fig. 4). The slope analysis in the ANCOVA showed that oxyurid relative abundance differed significantly across habitats with more rare species at low habitat loss ($F_{2, 33} = 28.28$, $P < 0.001$). Furthermore, the slope at low habitat loss is significantly steeper than the slopes at intermediate and high habitat loss, as shown by the *habitat loss level x oxyurid species rank* interaction term ($F_{2, 33} = 13.93$, $P < 0.001$), which indicates that oxyurid species are more evenly distributed in the landscapes with intermediate and high habitat loss than in the landscapes at low habitat loss (Fig. 3).

## Network structure

The size of our networks (hereafter referred to as tortoise–oxyurid networks) was similar in the number of oxyurid species among landscapes, suggesting that *T. graeca* maintains a

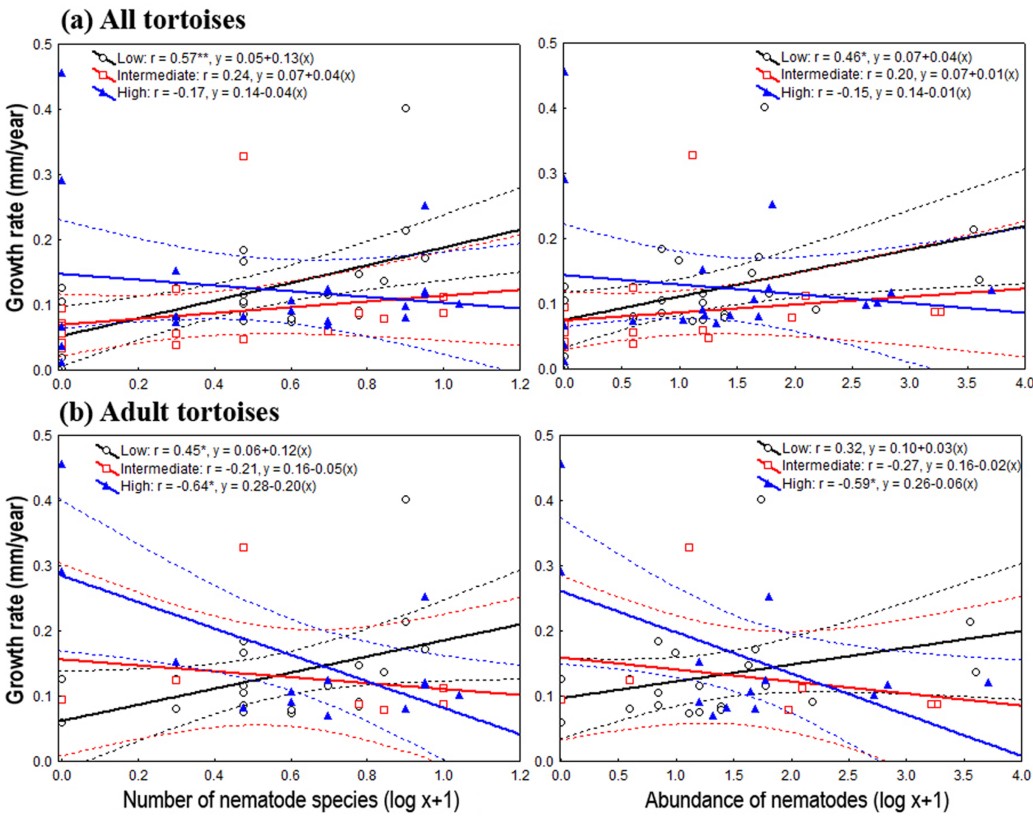

**Figure 2 Relationship between oxyurid nematode species richness and abundance and tortoises' growth rates.** Best-fit linear regressions, with 95 % confidence intervals, between oxyurid nematode infestation (i.e., species richness and abundance) and tortoises' (*Testudo graeca*) growth rates in scrubland landscapes in southern Spain. Tortoises were present in landscapes sustaining different levels of habitat loss (low, intermediate and high). (A and B) Tortoise populations at all life stages. (C and D) Adult tortoise populations only. Significant relationships are indicated by *$P < 0.01$ an/or **$P < 0.001$. The values for the linear regression equation ($y = a + bx$) and the resulting *r*-values are indicated for each case.               

stable community of these oxyurids at the three levels of habitat loss (Table 2; Fig. 4). The lowest diversity of interactions (*ID*), indicating lower community stability, was at low habitat loss. In contrast, the intermediate level of habitat loss showed the greatest specialization level ($H_2'$), implying that individuals were infested by fewer oxyurids in these landscapes, whereas individuals were the most generalized (i.e., showing higher functional redundancy) at high habitat loss. This agrees with the greater connectance in such habitats, reflecting a more uniform spread of interactions among oxyurid species in the community at high habitat loss (Table 2; Fig. 4). The three levels of habitat loss showed a significant nested pattern of interactions, though the landscapes with intermediate habitat loss exhibited the greatest nestedness values (Table 2). Furthermore, at the individual network level, body traits and growth rates explained to some extent network parameters (Table 2). Consistently in all habitats, heavier and larger tortoises showed a significantly positive relationship with species-degree, that is, with species richness of oxyurids in an individual, but not with species-specialization (*d'*). Moreover, species-degree tended to increase with
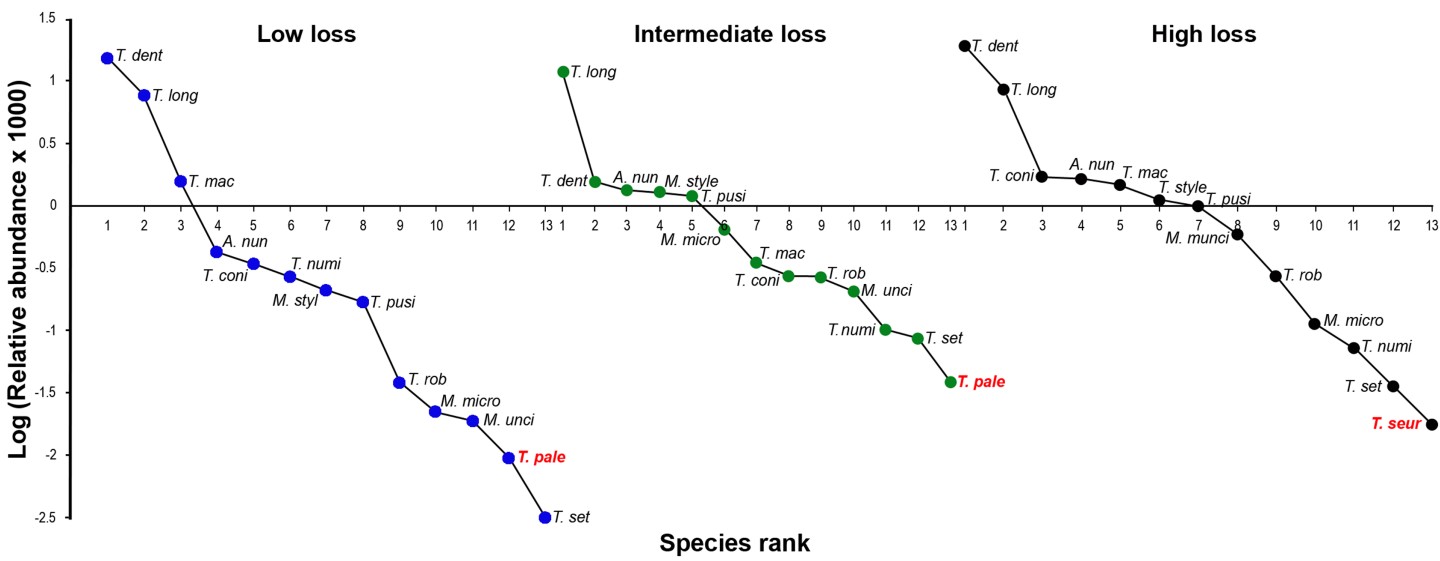

**Figure 3 Rank/abundance oxyurid nematode species plots.** Rank/abundance plots for the oxyurid nematode community infesting tortoises (*Testudo graeca*) in scrubland landscapes with different levels of habitat loss (low, intermediate and high) in southern Spain. For each landscape, the relative abundance of each nematode species is plotted on a logarithmic scale against the species' rank, from the most to the least abundant species. Nematodes included the following species: *Alaeuris numidica* (*A. num*), *Mehdiella stylosa* (*M. styl*), *M. uncinata* (*M. unci*), *M. microstoma* (*M. micro*), *Tachygonetria longicollis* (*T. long*), *T. dentata* (*T. dent*), *T. conica* (*T. coni*), *T. robusta* (*T. rob*), *T. macrolaimus* (*T. mac*), *T. numidica* (*T. numi*), *T. setosa* (*T. set*), *T. palearticus* (*T. pale*), *T. pusilla* (*T. pusi*), and *T. seurati* (*T. seur*). Nematode species not found under all habitat loss conditions are in bold red.

both tortoise weight and carapace length in low (weight, $\chi^2$ = 5.76, df = 1, $P$ = 0.016; length, $\chi^2$ = 6.01, df = 1, $P$ = 0.014), intermediate (weight, $\chi^2$ = 6.88, df = 1, $P$ = 0.0087; length, $\chi^2$ = 13.42, df = 1, $P$ = 0.0002) and high habitat loss (weight, $\chi^2$ = 4.33, df = 1, $P$ = 0.037). In this case, carapace length was positively related to species degree, though not significantly. In contrast, species-degree was related to individual growth rates only in landscapes with low levels of habitat loss ($\chi^2$ = 6.65, df = 1, $P$ = 0.01), indicating that tortoises that grow faster interact with more oxyurid species in this habitat.

## DISCUSSION

Overall, the level of habitat loss and the conveyed environmental and land use changes were found to affect some aspects involved in the ecological interactions (e.g., at the population and individual level) between the spur-thighed tortoises and their adult hindgut oxyurids. Oxyurid infestation showed differential association only with hosts' growth rates, across levels of habitat loss (*Amo, López & Martín, 2007*; *Janin, Léna & Joly, 2011*; *Benavides et al., 2012*). A possible explanation is that body weight and carapace length are strongly correlated to tortoises' age (*Rodríguez-Caro et al., 2013*), and age classes were not homogeneously distributed across landscapes, which lead to great variability among sites. Conversely, individual growth rate depends on the relation between age and size and therefore is not affected by age distribution among populations (Appendix 1). Furthermore, larger and heavier tortoises were infected by more oxyurid species. These changes were evident in spite of the similar oxyurid communities in terms of species richness and composition across landscapes. In general, the association of oxyurid

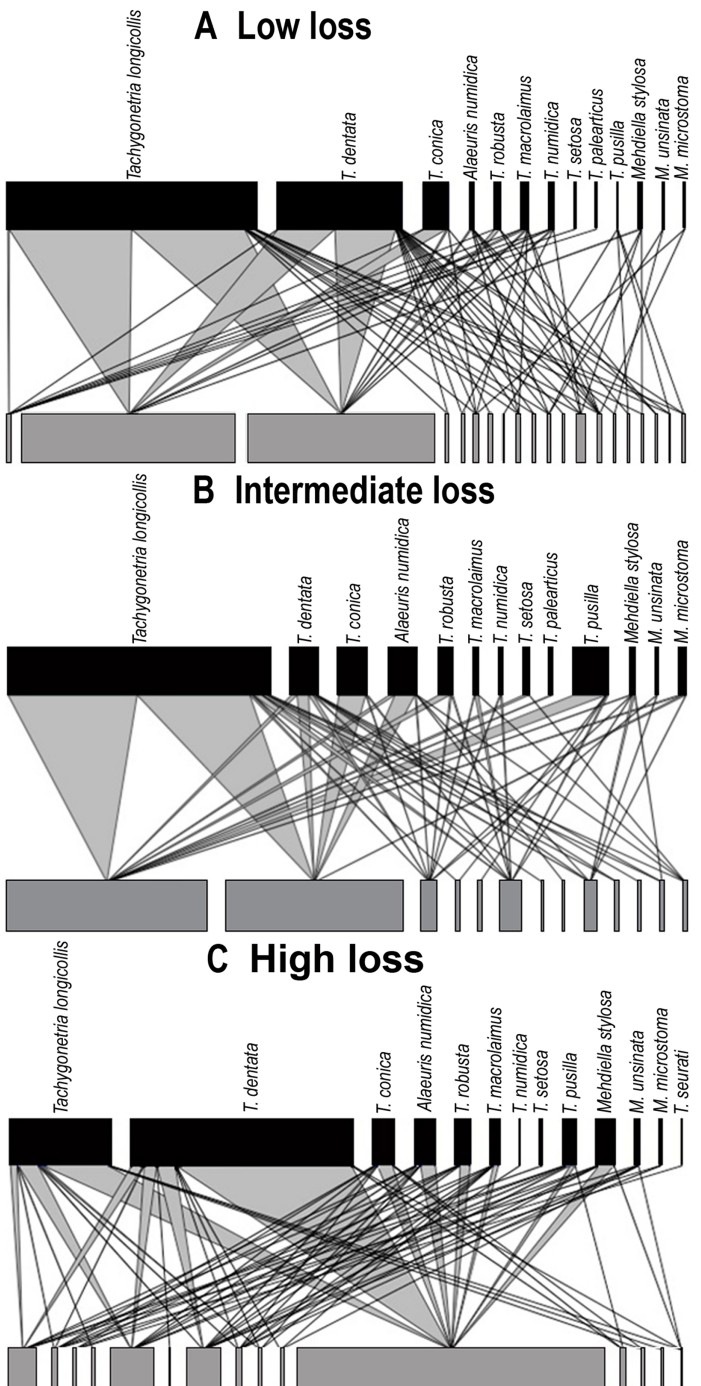

**Figure 4 Quantitative bipartite plots of the tortoise–oxyurid species ecological networks encompassing different levels of habitat loss.** Quantitative bipartite plots of the tortoise–oxyurid species ecological networks for each landscape encompassing different levels of habitat loss (low, intermediate and high) in southern Spain. (A) Tortoise–oxyurid ecological network at low habitat loss. (B) Tortoise–oxyurid ecological network at intermediate habitat loss. (C) Tortoise–oxyurid ecological network at high habitat loss. The bar size represents the number of interactions each oxyurid species (black) had with each individual tortoise (*Testudo graeca*) in the population (grey), and the linkage width represents the proportion of oxyurid individuals involved in such interactions.

**Table 2 Tortoise–oxyurid network parameters.** Tortoise–oxyurid ecological network attributes at two hierarchical levels in landscapes that differ in their levels of habitat loss in scrublands of southern Spain.

| Hierarchical level | Metrics | Low | Intermediate | High |
|---|---|---|---|---|
| Network level | | | | |
| | No. of tortoises | 20 | 13 | 15 |
| | Oxyurid species | 13 | 13 | 13 |
| | Nestedness ($WNODF$) | 37.23 | 45.90 | 43.696 |
| | Connectance ($C$) | 0.269 | 0.308 | 0.364 |
| | Interaction diversity ($ID$) | 2.034 | 2.371 | 2.397 |
| | Specialization index ($H_2'$) | 0.125 | 0.335 | 0.141 |
| Individual level (mean ± SE) | | | | |
| | Specialization index ($d'$) | 0.228 ± 0.050 | 0.187 ± 0.031 | 0.188 ± 0.027 |
| | Number of interactions ($degree$) | 3.50 ± 0.494 | 4.00 ± 0.828 | 4.73 ± 0.777 |

infestation with growth rates was related to the level of habitat loss, ranging from positive, to neutral, to negative, for landscapes encompassing low, intermediate and high habitat loss, respectively (*Amo, López & Martín, 2007*; *Janin, Léna & Joly, 2011*).

## Habitat loss and the costs and benefits of oxyurid nematode infestation

The modification of the original scrubland vegetation by anthropogenic activities produces strong shifts in the spatial configuration of the natural landscapes, as well as on their physical (e.g., rainfall and temperature) and biological environments (e.g., vegetation structure, species diversity and resources availability) (*Patz et al., 2004*; *Qian & Ricklefs, 2006*; *Amo, López & Martín, 2007*; *Janin, Léna & Joly, 2011*; *Suzán et al., 2012*; *Pringle, 2016*). These environmental shifts may occur at multiple spatial and temporal scales, which affect complex species interactions, including host-oxyurid interactions of any kind (e.g., positive, negative and/or neutral) in the case of herbivorous reptiles (*Harvell et al., 2002*; *Amo, López & Martín, 2007*; *Tylianakis, Tscharntke & Lewis, 2007*; *Janin, Léna & Joly, 2011*; *Benavides et al., 2012*; *Benítez-Malvido et al., 2016*; *Benítez-Malvido, Lázaro & Ferraz, 2018*; *Hernández-Martínez et al., 2019*). Although our findings represent the standing incidence of adult oxyurids in *T. graeca* tortoises (i.e., a single observation in time), we detected that changes in landscape configuration, exemplified by the amount of suitable habitat, influenced some aspects of host-oxyurid interactions as revealed by differences in oxyurid infestation levels, oxyurid species ecological evenness, tortoises' growth rates and by differences in the networks' structure. There is evidence showing that the most disturbed and homogenized landscapes (e.g., highly degraded landscapes) are more susceptible to pest and pathogen invasions because human activities facilitate the movement of exotic/invasive species to new areas and provide the conditions for their establishment. An additional reason is that hosts might change their behavioral and physiological ecology (*Patz et al., 2004*; *Qian & Ricklefs, 2006*; *Amo, López & Martín, 2007*; *Janin, Léna & Joly, 2011*; *Tompkins et al., 2011*; *Suzán et al., 2012*). Another factor

influencing pathogen transmission and infestation, not detected in the present study, is host population density (*Arneberg, 2002*). Therefore, the type of endosymbiont interaction (positive, negative and/or neutral) between tortoises and hindgut oxyurids seems to depend on the environmental context, in which differences in resource availability (e.g., host food type and habitat structure) across habitats could be driving the observed variations among them (*Tompkins et al., 2011*; *Pringle, 2016*). Large oxyurid infestation was negatively associated with growth rate for adult tortoises in landscapes at high habitat loss, suggesting a parasitic interaction (Fig. 3). Only under unusual circumstances, such as reptiles kept in captivity, are oxyurids pathogenic (*Jacobson, 2007*; *Chávarri et al., 2012*). High habitat loss and fragmentation (extreme conditions) might have lowered the capacity of tortoises to cope with a parasite infection in more challenging conditions; alternatively, those individuals with low growth rates might have poorer acquired immunity to oxyurids (*Amo, López & Martín, 2007*; *Janin, Léna & Joly, 2011*). Recording oxyurids at all life stages in feces is indicative that oxyurids have been inhabiting tortoises' hindguts long enough to complete their life cycles, and therefore, probably affecting some aspects of tortoises' fitness (*Jacobson, 2007*). Our results also showed, however, that high oxyurid species richness and abundance in tortoises are positively related with growth rates at low habitat loss, which suggests that oxyurid infestation could play a positive role in the digestion and thus assimilation of plant matter, possibly indicating a mutualistic interaction (*Roca, 1999*; *Mitchell & Diaz-Figueroa, 2005*; *O'Grady et al., 2005*; *O'Grady & Dearing, 2006*; *Jacobson, 2007*). Furthermore, at low habitat loss, tortoises could have a greater foraging range, that is, could show higher resource seeking activity enhancing a faster growth though simultaneously being exposed to greater oxyurid infection, regardless of their parasitic/commensal/mutualistic nature (*Amo, López & Martín, 2007*; *Janin, Léna & Joly, 2011*). Finally, tortoises within landscapes at intermediate habitat loss showed no significant relationship between oxyurid infestation and growth rates, suggesting a commensal interaction in which oxyurids had no apparent influence on tortoises' growth. Alternatively, tortoise hosts in these landscapes might be able to sufficiently allocate resources for growth and parasite resistance so that parasite acquisition and resistance balanced out.

In contrast to what might be expected, the ANCOVA showed that overall growth rates were greater at high habitat loss for adult tortoises than at low habitat loss. In this context, some native animal species are able to persist in degraded lands through morphological, physiological and behavioral adjustments (*Isabirye-Basuta & Lwanga, 2008*; *Tompkins et al., 2011*). Terrestrial tortoises are ectotherms and therefore may strongly react to environmental changes (*Kaspari & Valone, 2002*; *Anadón et al., 2006*). It is possible that reduced scrubland vegetation cover (greater canopy openness) in landscapes with high habitat loss aid tortoises in more effectively regulating their body temperatures (*Row & Blouin-Demers, 2006*; *Poulin & George-Nascimento, 2007*; *Sato et al., 2014*). In addition, in some reptile species, an adequate thermoregulatory behavior has been found to optimize the digestive performance, which in turn can affect growth rates (*Avery et al., 1993*; *Anguilletta, Hill & Robson, 2002*). Further studies should consider other indicators of

habitat loss on the resource availability for tortoises and the impact on their populations, such as survival and reproductive success (*Rodríguez-Caro et al., 2013*).

For the present study region, habitat loss and fragmentation have been found to negatively affect the persistence of tortoise populations, but also the presence of native scrubland habitat in the vegetation matrix has been reported to enable *T. graeca* to persist in anthropogenic landscapes (*Anadón et al., 2006*). There are no base line studies, however, that help us to precisely determine the overall impact of land-use change in *T. graeca* populations because this region has been modified for many centuries by anthropogenic activities of distinct magnitude and intensity (*Hewitt, 2011*). Therefore, long-term studies on *T. graeca* population dynamics and oxyurid infestation dynamics and their consequences (in survival, growth rates and reproduction) are crucial for understanding habitat loss processes in population's viability, as well as the wide-ranging role of oxyurid endosymbiotic interactions in tortoises.

Furthermore, it has been indicated that long-term studies are crucial for understanding the impact of sporadic natural disturbances and/or catastrophic events (e.g., flooding, fire and drought) on disease emergence, virulence and transmission (e.g., infection cycles) as well as on overall host health (*Beldomenico & Begon, 2009*; *Tompkins et al., 2011*; *Zarazúa-Carbajal et al., 2016*; *Hernández-Martínez et al., 2019*). The latter might apply as well for nematode endosymbionts. This information is relevant for the management of wild *T. graeca* populations and cannot be obtained in short-term studies, such as the present one. For instance, the information gathered in the present study was obtained during the spring months when resource availability is high (i.e., with several plant species blooming and fruiting). There is no information, however, on the prevalence and intensity of oxyurid infestation during seasons when food sources are scarce, nor between years, nor on the consequences for tortoise populations (*Tompkins et al., 2011*; *Benavides et al., 2012*).

## Ecological networks in the management of spur-thighed tortoise populations

The oxyurid community infesting *T. graeca* populations was not randomly assembled but nested, revealing a structured pattern for all levels of habitat loss. Tortoise–oxyurid network properties appear to be maintained in landscapes with contrasting levels of habitat loss, despite differences in species dominance, as well as host number and quality (tortoises' sex, age and size). Our study contributes to the understanding of the structure of ecological interactions in contrasting landscapes. Within the same host population, we can often find both more selective (those individuals colonized by a few oxyurid species) or more opportunistic (those individuals colonized by many oxyurid species) individuals. Some network distribution metrics, such as $ID$ and $H_2'$, may be suitable indicators for detecting human-induced changes in some endosymbiotic interactions (e.g., mutualistic, commensal and parasitic), including tortoise–oxyurid interactions (*Kareiva, 1987*; *Tylianakis, Tscharntke & Lewis, 2007*; *Kaiser-Bunbury & Blüthgen, 2015*). For instance, theory predicts positive correlations between host population density and/or body mass with parasite species richness (*Morand & Poulin, 1998*; *Arneberg, 2002*). This relationship

could only be detected at the individual network level where tortoise body mass (i.e., weight) was associated with the number of oxyurid species interactions.

We found that as habitat loss increases host specificity decreases. In other words, hosts are infested by a wider array of oxyurids, making tortoises more vulnerable in an already resource-limited environment in case the oxyurid infestation turns out to be parasitic. The greater nested pattern in landscapes with intermediate and high habitat loss may also imply that oxyurids can be more easily transmitted through the entire network, something that would be less likely if the network had a more modular structure (i.e., consisting of very specialized host-oxyurid interactions).

It is worth noting, however, that the lack of identification of oxyurid eggs and larvae could have affected our results in several ways. For instance, oxyurid species richness might be underestimated and, thus, some network parameters might have been misinterpreted. The use of adult oxyurids, however, made our study more conservative, as it was also inconvenient considering all oxyurid eggs and larvae as "single species" in our ecological networks. Furthermore, the positive host-parasite-nutrition relationships might be stronger in adult oxyurids given their more developed gut microbial community; whereas adult oxyurids might, as well, be more pathogenic than eggs and larvae (*Jacobson, 2007*).

## CONCLUSIONS

We aimed to understand the ways habitat loss affected spur-thighed tortoise populations and to link habitat loss and tortoise–oxyurid interactions by using a network approach. Our findings indicate that the level of habitat loss is associated with growth rates and host-endosymbiont interactions; in this sense, this study is the first to provide the structure of such networks. Animals, at any stage of their life cycles, are subject to infestation by many different organisms that may modify, improve or interrupt the vital functions that affect their fitness (*Patz et al., 2004*; *Aguirre & Tabor, 2008*; *Suzán et al., 2012*; *Pringle, 2016*). Moreover, molecular studies-based on barcoding-would be very useful for identifying eggs and larvae and therefore, to completely describe the oxyurid species conforming the nematode community in the spur-thighed tortoise hindgut (*Jacobson, 2007*; *Origgi & Paré, 2007*). On the one hand, the rapid degradation of natural habitats might cause a large proportion of species and populations to be vulnerable to disease, and native and exotic pathogens could become a threat for both plant and animal communities (*Anderson et al., 2004*; *Patz et al., 2004*; *Vurro, Bonciani & Vannacci, 2010*; *Tompkins et al., 2011*; *Benítez-Malvido, Lázaro & Ferraz, 2018*). On the other hand, habitat degradation might also cause the loss of mutualistic and commensal interactions. Therefore, it is essential to understand the nature and dynamics of symbiotic interactions, the life histories of the animal hosts and the effects of the local environmental conditions on all of them.

## ACKNOWLEDGEMENTS

We warmly thank J. M. Lobato for his careful technical support and to P.M. Mojica and M. Fernández for their help with sampling tortoises. We are also very grateful to

M. Chávarri for the isolation and identification of oxyurid species. An anonymous reviewer made very valuable suggestions to previous versions of the manuscript.

### Funding

This manuscript was partially written while Julieta Benítez-Malvido was on sabbatical at the Institut Mediterrani d'Estudis Avançats, and Héctor Hugo Siliceo-Cantero on a post-doctoral scholarship, both supported by the Dirección General de Asuntos del Personal Académico-Universidad Nacional Autónoma de México. Financial support was granted by the Spanish Ministry of Economy, Industry and Competitiveness (Nos. CGL2012-33536, CGL2015-64144, CGL2017-88122-P), by the European Regional Development Fund (No. MINECO/FEDER, UE) and by the Regional Valencian Government (No. GV/2019/039) The funders had no role in study design, data collection and analysis, decision to publish, or preparation of the manuscript.

### Grant Disclosures

The following grant information was disclosed by the authors:
This manuscript was partially written while Julieta Benítez-Malvido was on sabbatical at the Institut Mediterrani d'Estudis Avançats, and Héctor Hugo Siliceo-Cantero on a post-doctoral scholarship, both supported by the Dirección General de Asuntos del Personal Académico-Universidad Nacional Autónoma de México.
Spanish Ministry of Economy, Industry and Competitiveness: CGL2012-33536, CGL2015-64144, CGL2017-88122-P.
European Regional Development Fund: MINECO/FEDER, UE.
Regional Valencian Government: GV/2019/039.

### Competing Interests

Anna Traveset was an Academic Editor for PeerJ.

### Author Contributions

- Julieta Benítez-Malvido conceived and designed the experiments, analyzed the data, prepared figures and/or tables, authored or reviewed drafts of the paper, approved the final draft.
- Andrés Giménez conceived and designed the experiments, performed the experiments, contributed reagents/materials/analysis tools, authored or reviewed drafts of the paper, approved the final draft.
- Eva Graciá conceived and designed the experiments, performed the experiments, contributed reagents/materials/analysis tools, authored or reviewed drafts of the paper, approved the final draft.
- Roberto Carlos Rodríguez-Caro conceived and designed the experiments, performed the experiments, analyzed the data, contributed reagents/materials/analysis tools, prepared figures and/or tables, approved the final draft.
- Rocío Ruiz de Ybáñez conceived and designed the experiments, performed the experiments, contributed reagents/materials/analysis tools, authored or reviewed drafts of the paper, approved the final draft.
- Héctor Hugo Siliceo-Cantero conceived and designed the experiments, analyzed the data, prepared figures and/or tables, approved the final draft.
- Anna Traveset conceived and designed the experiments, analyzed the data, prepared figures and/or tables, authored or reviewed drafts of the paper, approved the final draft.

## Animal Ethics

The following information was supplied relating to ethical approvals (i.e., approving body and any reference numbers):

The Dirección General de Gestión del Medio Natural de la Junta de Andalucía (SGB/FOA/AFR) and the Delegación General de Medio Natural de la Comunidad Autónoma de la Región de Murcia (AUT/ET/UND/48/2010) granted permission to sample the tortoises and nematodes.

## Field Study Permissions

The following information was supplied relating to field study approvals (i.e., approving body and any reference numbers):

Part of this work was carried in public mountains. There, our work in public mountain areas was undertaken with permission of the Regional Governments of Murcia and Andalusia (in accordance with article 33 of the Spanish Constitution of 1978). The NGO Global Nature gave us their express authorization to carry out work at their private properties. We also worked in private forested areas that were not fenced or otherwise indicated restricted access. In Spain, where access is not expressly prohibited, permission to enter forested areas is allowed (even if they are private areas) in accordance with the law of mountains (law 21/2015, which modifies the law 43/2003). Where it was possible, we obtained verbal permissions from the owners of private forested areas.

## Data Availability

The raw measurements are available in the Supplemental Files.

## Supplemental Information

Supplemental information for this article can be found online at http://dx.doi.org/10.7717/peerj.8076#supplemental-information.

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
