# Peer review of "Impact of habitat loss on the diversity and structure of ecological networks between oxyurid nematodes and spur-thighed tortoises (Testudo graeca L.)"

_PeerJ, doi:10.7717/peerj.8076_

## Round 0.1 · original submission · Major Revisions

I found this to be very interesting study with a novel angle. The reviewers were very diligent and have made excellent and constructive comments leaving little for me to add. The choice and limitations of the parasitological methods, and of which subset of the data to analyze, as well as a full consideration of their implications, is the most important issue to address. I would like to point out that the original data needs to be included in the supplement, this seems to be only partially the case.

Reviewer 1 ·

Basic reporting

• The language is primarily clear and unambiguous, and professional English is used throughout. There are some parts where further clarification is needed, and a few typos remain. Minor English language finetuning is needed e.g. in lines 121-122; 208-209; 262. Throughout, the potentially inappropriate use of causal language should be amended where there is no clear causal relationship. E.g. L291 “consequences for”; L301 “significantly affected” refer to the association between nematode presence and growth rate, where the direction of the causality is likely, if anything, the opposite to that inferred in text.
• The introduction is reasonably successful in demonstrating how the work fits into the broader field of knowledge, but the consideration of prior literature could be more balanced.
• The structure of the paper is clear and easy to follow. The figures and tables are legible and appropriate, but lack some axis titles and legend detail.
• Partial raw data has been shared, but a large proportion of essential information is missing.
• The submission appears to be self-contained, but there is some mismatch between the hypotheses and results. Some relevant methodological details and results are apparently missing or unclear.

Detailed comments
The paper deals with species interactions and the way in which habitat degradation or loss influences ecological networks centered around a specific species of interest, a severely threatened tortoise. Considering ecological networks, with a focus on the effects of the environment on species communities and interactions, have been largely overlooked in conservation studies to date. The study is, therefore, well-motivated and uses a potentially interesting network approach to the topic. It is also generally fluently written and well-structured. However, I believe there are several quite problematic points in the methods and the interpretation of the results that render the study outcomes unreliable. These points must be justified and clarified for the paper to meet publication standard.

Introduction
The introduction reads well, makes interesting points, and describes the aims of the present study. The growth rate component could be better linked to prior literature and the study hypotheses, however. For example, it would be useful to add something about what factors are known to influence (tortoise) growth rate and through what mechanisms habitat loss and nematode infections should influence growth (see e.g. references below). Additionally, any information on the presumed mechanism mediating nematode abundance and richness effects on host health might be useful for making predictions about their associations.

I would also appreciate a little more information on the proposition in L 71-73:
“for herbivorous reptiles, hindgut nematodes might be parasitic but might also play a positive role in the digestion and assimilation of plant matter”
I believe this is quite an unusual suggestion for many researchers who generally perceive gut nematodes as parasitic throughout, and seen as this is an important part of the hypotheses tested, I would appreciate some additional support for the idea. E.g. it seems to me likely that beneficial effects of gut nematodes might operate (at least in part) via nematode-microbiota interactions in the gut. I missed some background on the hypothesized association between host growth and parasite infections, which is integral to the study.

Experimental design

• The research question is provided and is meaningful, but could be clarified further. The study identifies a knowledge gap, but it is not entirely clear that the study is appropriately designed to fill that gap. A fundamental issue relevant to the collection of relevant data (the parasite life cycle) appears to have been overlooked, as the analyses are based on the prevalence of adult nematodes in fecal samples, instead of the more common (though also flawed) method of examining nematode eggs, or molecular identification methods. While some useful information could, nevertheless, be gained from the study on the prevalence of certain parasite species in different habitats, the measure may come with a high measurement error and risk of non-detection, rendering the results potentially quite unreliable. The potential consequences of this choice of method on the outcomes have not been, but should be discussed.
• It is somewhat unclear whether the investigation was conducted rigorously and to a high technical standard, because the data collection methods are insufficiently described. There are some points of potential concern in the data analyses (see below).
• Ethical standards in the field appear to have been followed appropriately.
• The methods were not described with sufficient information to be reproducible by another investigator. For example, the measurement of individual growth rate (the primary outcome variable) has not been described. Furthermore, no information is provided about the treatment and analysis of the parasite samples (basis for the primary predictor variable). See details below.


Detailed comments
Information on the habitat, study species, and aspects of the parasites are appropriately described and relevant, but there are a few important details that should be clarified.

L184-188: given the importance of the growth metric for the outcomes of the study, I would like for more details here rather than referring to the Rodríguez-Caro et al. (2013) paper for all details. If I had to guess what this means, I would assume that growth measured using recapture data from change in size over these 3 months, but surely this would not be sufficient time to record growth in a meaningful way? Rodríguez-Caro et al. appear to have used repeated measures of the same individuals over a 4-yr period to derive the growth curves. Is the population used for this study long-term monitored? Please provide details on the measurements and growth rate calculation used in the present study.
If the growth estimates are derived from longitudinal data, how relevant is this metric for the point sampling of the nematode community, and would the causality relationship change? E.g. statement in L 300-301 “both the number of species and nematode abundance significantly affected growth rates”.

L201: ditto, at least a brief description of the parasite sample processing and data collection is needed here.

L205-207: Please specify which of these species belong to Oxyurida and Ascaridida, or other orders. This is relevant for the rest of the paragraph, where you focus on these two orders.

L207-209: I find the decision to exclude eggs a potentially very problematic issue, given the life cycles of these nematodes. At least in mammals, intact adult oxyurids should only be found in feces under very heavy infestation or possibly during severe gastrointestinal distress (e.g. prolonged diarrhea), whereas the eggs are actively deposited in the perineum. As adult oxyurids females migrate to the perineum, dead adult worms might occasionally be discovered in feces, but this is generally a very unreliable way of diagnosing an ongoing infection. Oxyurid infections are, therefore, quite difficult to assess reliably without the “perianal sticky-tape test” for eggs. Adult ascarid worms should be found in feces even less often because they live in the small intestine and dead worms should generally be broken down in digestion, whereas eggs are actively deposited to be dispersed in feces.
While I understand that the eggs cannot be visually identified to species level and therefore cannot be used to assess species abundance/diversity, the choice to focus on adults may severely bias the results. The potential consequences of this decision for the results and conclusions should, therefore, be clearly outlined and justified, and the findings discussed strictly in the light of these restrictions.
For example, how does the choice of method relate to the relative apparent abundance/diversity of oxyurids relative to other nematodes? How would this be reflected in the rank-abundance plots? And what would it do to the nestedness/ community analyses if a large number of actually prevalent species are not observed?
There may be no reason to assume that the adult worm shedding should vary according to habitat disturbance, but if only a small proportion of nematode species are found in feces in the adult stage, it may be practically impossible to say anything about the actual nematode communities or differences in those communities among sites.
Additionally, statements such as L 317-318 “dominant nematodes infecting tortoises throughout the landscapes were T. dentata and T. longicollis” should be amended throughout to acknowledge this constraint (i.e., here, the adult nematodes most commonly found in the feces were of the species…).

L 226-229: “Body weight and carapace length are strongly correlated to tortoise´s age (Rodríguez-Caro et al. 2013), whereas age classes were not homogeneously distributed across landscapes; therefore, weight and carapace length were not used further in the analyses related to the impact of nematode infestation on body traits across habitat types.”
The rationale of this decision is not obvious to me – why not include these traits as covariates in the analysis instead? Also, please remind the reader which body traits are considered if not the weight and length; I am a little confused at this point as the body measures, as far as I have understood, were L 183-185: “age estimated from the carapace growth ring (Rodríguez-Caro et al., 2015), body weight (g) and carapace length”. This seems to leave nothing to associate the nematode infections with. Please clarify this.

Please clarify whether tortoises were sampled multiple times each, and if so, how you treated the repeated measures in each of the analyses. Did you select one sampling event to assess parasite load or pool several samples per individual?

Given the relatively long sampling window, it would be useful to test and mention whether the nematode community changed over the season, and whether this might influence your network analyses.

L236-237: Please clarify why these were log-transformed. Note also that both measures may be considered as counts, which should ideally not be log-transformed. See e.g. O’hara, Robert B., and D. Johan Kotze. "Do not log‐transform count data." Methods in Ecology and Evolution 1.2 (2010): 118-122.

L245-246: I would suggest reconsidering the wording here to make this easier to grasp, and use careful and consistent wording throughout this sentence (and later in the paper where talking about this concept). Please clarify what this refers to - the relative abundance of each nematode species (where - per individual, site, or entire sample?) against the ranked abundance of those same species (where?)? This sounds initially as if the relative abundance was plotted against itself, and the absence of axis titles in the figure doesn’t help comprehension. I continue to have trouble with this sentence, though going over the paragraph multiple times (I think) I managed figure out what this means.

L277-285: Please indicate why you calculated each of these metrics (especially what additional information was to be gained from each of them)

L279: Please clarify what exactly “number of cells in the matrix” and “interaction diversity” refer to in this context

L285-287: Please remove repetition here

Validity of the findings

• Based on the omissions in the method description, I am not fully convinced that the data are robust, statistically sound, and controlled. Only partial data used in the analyses have been made available.
• The conclusion are linked to original research question, but in my opinion they occasionally venture quite far beyond the supporting results, e.g. L382-384. Speculation has not always been identified as such. See details below.

Detailed comments

Results
It would also be useful to have some more information on the tortoises that presented with no adult nematodes, relative to those that had them. Were they more commonly found in one or the other habitat type, did their growth rates differ?

Please tone down language implying causal relationship between growth rates and nematode load. Instead of “consequences”, effects”, I suggest using “association” or “relationship”. Appropriate reporting is already seen in e.g. L311-313.

L294: presumably this should be between 0 and 10 species (rather than 1 - 10)

L308-309: These statistics (richness, abundance) do not seem to be the correct ones for this sentence, please check.

L346: What does this antagonistic interaction refer to? Antagonism between the different nematode species in the community (as opposed to antagonism between host and nematodes)? It would be interesting to explain this further in the discussion.

Figure 2: Indicate whether this is excluding those tortoises that had zero nematodes. I believe it would be useful to include those individuals here for completeness.
Please clarify what the numbers in the figure panels after the r-coefficient refer to, i.e. what does this sequence mean: y= … - …(x)

Figure 3: bolding is not clear from figure - consider adding a color code. The axes are not titled, please amend.

Table 1 legend: “oh” -> “of”.
Does “density of nematodes” mean abundance? Please use uniform terminology in text and tables.


Discussion
L349-355: In light of the constraints mentioned above, I would be willing to cautiously trust the finding of similar nematode communities (of those species that can be observed as adults in feces) among habitat types. For the rest of the paragraph (tortoise growth – habitat – nematode community associations), it seems to me unlikely that this is the most parsimonious explanation for the observed associations. It seems to me substantially more probable, that the habitat disturbance alters the food availability, stress, movement ranges or other factors that influence both the growth of the animals, and consequently their exposure/susceptibility to nematode infections. Independently of growth, nematode abundance and diversity in populations may also be influenced by features of the environment, including population density and frequency of interactions e.g. through patchy resource distribution. These alternative explanations cannot be distinguished based on the presented data. I suggest refraining from drawing strong causal conclusions, such as using the word “impact” when describing the observed associations.

I agree that it is an interesting possibility that certain nematodes could promote the subsistence of their host species by facilitating food digestibility. However, it may be premature to draw such a conclusion from the present study, and therefore, I suggest toning down the discussion around the “symbiotic” relationship of nematodes with tortoises, which is carried through much of the discussion.
E.g. L 382-384: This seems to me to be a long leap of faith given the available data, while there are other, perhaps better supported potential explanations.

First, an apparent positive effects of nematode richness on growth could also arise through the influence of a shared third variable, such as a more varied diet. Second, there is a well-known association between host size and parasite load, which might explain at least part of your findings. Third, you may want to consider (and should at least mention) the more classical explanation: that the intestinal communities of organisms are determined via the patterns of acquisition from the environment, and the capacity of the host to get rid of unwanted organisms. For example, using the statements from the abstract:
1) “At low habitat loss, nematode infestation was positively associated with growth rates (suggesting a mutualistic nematode-tortoise relationship)”, could alternatively be explained through higher resource seeking activity such as broader habitat use by those individuals that facilitated a faster growth but simultaneously exposed the individual to more nematode eggs, regardless of their parasitic/commensal/symbiotic nature.
2) “The association became negative at high habitat loss (suggesting a parasitic relationship)” could also be explained with a lowered capacity of individuals to cope with a parasite infection in more challenging conditions, or those individuals with low growth rates may have poorer acquired immunity to parasites etc.
3) “No relationship was observed when habitat loss was intermediate (suggesting a commensal relationship)” could also indicate that the hosts were able to sufficiently allocate resources to growth and parasite resistance that parasite acquisition and resistance balanced out.

This is only to point out, that there are alternative, previously supported hypotheses that could explain the observed patterns, even if all nematodes were parasitic. I suggest including these potential alternative explanations in your discussion and evaluating the support for each alternative with your results and the literature, perhaps suggesting further studies that could be used to distinguish between the potential explanations.


Data
Please also provide the tortoise body measures, growth rate, etc. that were used in the analyses in the data file.

Additional comments

Potentially useful references
Adamson, Martin. "Evolutionary patterns in life histories of Oxyurida." International Journal for Parasitology 24.8 (1994): 1167-1177.

Aho, John M. "Helminth communities of amphibians and reptiles: comparative approaches to understanding patterns and processes." Parasite communities: patterns and processes. Springer, Dordrecht, 1990. 157-195.

Arneberg, Per. "Host population density and body mass as determinants of species richness in parasite communities: comparative analyses of directly transmitted nematodes of mammals." Ecography 25.1 (2002): 88-94.

Benavides JA, Huchard E, Pettorelli N, King AJ, Brown ME, Archer CE, et al. From parasite encounter to infection: multiple-scale drivers of parasite richness in a wild social primate population. Am J Phys Anthropol. 2012;147:52–63.

Bordes F, Morand S, Kelt DA, Van Vuren DH. Home range and parasite diversity in mammals. Am Nat. 2009;173:467–74.

Coop, R. L., and I. Kyriazakis. "Nutrition–parasite interaction." Veterinary parasitology 84.3-4 (1999): 187-204.

Morand, Serge, and Robert Poulin. "Density, body mass and parasite species richness of terrestrial mammals." Evolutionary Ecology 12.6 (1998): 717-727.

Poulin, Robert, and Mario George-Nascimento. "The scaling of total parasite biomass with host body mass." International journal for parasitology 37.3-4 (2007): 359-364.

Stephenson, Lani S., Michael C. Latham, and E. A. Ottesen. "Malnutrition and parasitic helminth infections." Parasitology 121.S1 (2000): S23-S38.

Wilson K, Bjørnstad O, Dobson A, Merler S, Poglayen G, Randolph S, et al. Heterogeneities in macroparasite infections: patterns and processes. In: Hudson PJ, Rizzoli A, Grenfell BT, Heesterbeek H, Dobson AP, editors. The ecology of wildlife diseases. Oxford: Oxford University Press; 2002. p. 6–44.

Reviewer 2 ·

Basic reporting

This is a well-written and interesting study on the structure of nematode-tortoise networks and the structural changes within these networks with habitat loss. I particularly liked the point that the authors relate the degree of nematode infestation to tortoises’ growth rates, showing that the relationship changes from positive to neutral to negative with increasing habitat loss. This interesting finding indicates that changes in the interactions between nematodes and tortoise individuals with increasing habitat loss affect tortoise’ fitness.

This being said, I think the manuscript would be even stronger if the way network structure affects the fitness of tortoises was even more directly tested. At present, the authors analyze the interactions only at network level (e.g., with measures of nestedness or connectance). Reading the manuscript, I wondered if instead it would be helpful to analyze also species-level attributes of network structure. For example, they could test whether species degree (the number of nematodes affecting each tortoise), species strength or measures of species specialization (e.g., Blüthgen’s d’; a measure of how specialized a species or individual interacts within the community) can be directly related to the growth rates of the studied tortoise individuals. In my view, this would make an even stronger point that network structure directly relates to the fitness of interacting individuals, something which is indeed very rarely studied.

I provide more detailed and minor comments below, and hope they are helpful in further strengthening this interesting manuscript.

Abstract:
L41: Do you mean that rarity thus equals specialization? But this is not specialization measured at the species level but with network-level indices?
L43: In how far is connectance related to network size (e.g., the number of interacting species)? This two are usually strongly negatively correlated. Did you consider correcting for this, for instance using null model analysis? (see e.g., Dalsgaard et al. 2017, Ecography).

Introduction:
L57: This should be “Fahrig”.
L57: Note that Lenore Fahrig herself appears pretty critical about the very general statement that habitat fragmentation has primarily negative effects on biodiversity (see her recent interesting comment in Global Ecology and Biogeography 2019; 28:33-41).
L79: What kind of animal species?
L80-83: Also here, it would be interesting to know which species you are referring to.
L84ff: This paragraph comes rather abruptly. When reading the manuscript, I got the feeling it would be easier to understand your design if the region were first descript (this is the part from L111ff). You may consider moving this section before the part that starts in line 84. You could then finish the Introduction with the paragraph on network ecology (L95-110), before listing you hypothesis.
L122-123: It is unclear what you mean by “diverge with the level of habitat loss”, try to be more specific.

Methods:
L147-150: What is the scale at which you defined scrubland cover? For example, was there a predefined radius surrounding your study populations at which you conducted the landscape composition analysis?
L150-151: Where habitat loss, fragmentation and intensification related to each other, and could this possibly confound your results on the effects of habitat loss?
L154-160: I was very surprised to read that 76% landscape cover with suitable habitat is referred to as “high habitat loss” – to my naïve understanding, this would still look like a landscape with very high amounts of natural habitat remaining. I presume that tortoises may see this differently though, but please justify why a loss of 24% suitable habitat is already considered as the “high loss” category, and how this refers to tortoise biology.
L187: If space allows, shortly describe the Bertalanffy model for non-experts.
L208: Word missing between “because” and “the lack of”.
L209-219: These information would be better placed in the Introduction, to motivate the hypothesis of the study and explain why contrasting effects of nematode infection on growth rates can be expected.

Statistics:
In general I found the statistical analysis sound. However, I wondered if you could have used GLMs instead of log(x+1)-transforming the data (L236-237) (e.g., using a Poisson or negative-binomial distribution).

L256-258: How large was the proportion of individuals considered in the interaction network analysis?
L260-265: As far as I understand, these are then weighted interaction networks (not binary, which is great), perhaps include this point.
L268: If the networks are weighted, why not use the a weighted measure of nestedness such as wNODF?
L274: Please describe how ‘Null Model II’ works, for example, how is the community matrix randomized (e.g., with fixed connectance and/or fixed marginal sums)? This can be very important to the result (Dormann et al. 2017, Annual Review of Ecol. Evol. Syst.)
L274ff: Referring to the null model, this could also be used to standardize the network parameters, which may overcome artifacts by differences in network size across localities with different amount of habitat loss. (using a standardization such as SES = (obs_nestedness – mean(random_nestedness)) / sd(random_nestedness).
L277-280: As commented before, I would expect connectance and interaction diversity to be strongly correlated to network size.

Results:
L290: “some aspects” is vague, please be more specific.
L294: Provide means and a measure of spread (e.g. standard errors) throughout.
Figure 2: The symbols are pretty small and hard to discern (particularly when overlapping). If possible, increase their size or use colors to indicate the habitat types. Additionally, including confidence intervals of the regression fits would be helpful.
Figure 4: Please include a legend that displays how the number of interactions corresponds to the width of the boxes.

Discussion:
L370ff: But your landscape do not appear to be heavily modified in the sense that high amounts of natural/suitable habitat have been lost, when compared to most other studies on habitat loss.
L379-387: Here, more discussion and analysis how this contrasting infestation effects result would be great – possibly network analysis at the species-level could help here.
L400-404: These studies appear to contradict each other – how can any fragment regardless of its size support viable populations, and fragmentation still negatively affect them?
L405ff: Does this also mean that critical thresholds of habitat loss on tortoise survival remain unclear? Are there landscapes with higher amount of habitat loss than considered in your study in which large tortoise populations remain?
L430-435: See comment above why species-level indices may be even more informative.

Experimental design

see basic report above.

Validity of the findings

see basic report above.

Additional comments

see basic report above.

---

## Round 0.2 · Major Revisions

The revised version of the manuscipt has been improved, but a number of issues , regarding the interpretation of the data and the discussion of its limitations, remain to be addressed. The reviewer makes a convincing case that the assumptions made regarding the causal linkage between the detections of nematodes (based on one sampling occasion per individual) and the body traits and growth rates of the tortoises, are too strong, given what is known about the turnover of nematodes within indiciduals over time. This undermines the ability to satisfactorily address the second aim of the study as currently formulated. I recommend that the authors either drop that part or remove the causal interpretations. This may feel like watering down that part of the paper, but overall it would make it more robust.

Reviewer 1 ·

Basic reporting

Ok, see below

Experimental design

Ok, see below

Validity of the findings

I thank the authors for their efforts to clarify points that were unclear in the previous manuscript draft. They have successfully responded to most of my smaller comments and the added appendix and details on the measurement of growth rates significantly clarified the methods. Unfortunately, there has been little change to the two major points that I brought up in my previous review. Rather, the issues are more obvious to me now that the unclear points have been clarified.

FIRST, I'm sorry but I simply cannot buy into the suggested "effect" or “influence” of nematode loads on growth rates. Word choice aside, the assumption of an effect of current nematode loads on past tortoise growth is difficult to justify. The authors have not succeeded in convincing me that this is not a problem (although I appreciated the additions to L465-480, which helped balance the discussion!)
I ask the authors to please reconsider what is being implied by suggesting that certain nematode communities influence(d) the tortoises’ growth etc.. Unless I have completely misunderstood the way the study was conducted, this is not a trivial detail but has significant implications for the ecological relevance of the study. I’ll try to make my thinking clearer.
You are measuring adult Oxyurid nematodes in (adult) tortoises. You are associating this nematode load with growth rate over the past n years of the tortoise’s life. Over this time, the tortoise may or many not have been carrying certain species of nematodes. Nematode loads increase with age, as you point out, but there is also turnover in both the environment and within the host, so at times the nematode load may be higher, others lower, and the tortoises have likely gone through such phases in their lifetime. We cannot know the numbers or prevalences of those nematodes when the growth occurred, because a single sample was taken at the very end of the growth progression and nematode accumulation over the animal's lifespan to date.

How is it possible to infer a causal effect of this current nematode load on the growth of the animal at any time in the past? Simple logical reasoning of cause and effect suggests that a cause comes before effect, thus the causal relationship here could only be assumed to be that fast-growing animals accumulated and retained more nematodes over time (or that fast-growing animals more easily shed Oxyurids into feces when stressed by handling). The ecological implications including those for the host-nematode relationships are very different in these scenarios, and I think it is misleading to suggest any effect of nematodes on the host (especially the host's past life) based on the present study.

Given this problem, an attempt to infer the type of relationship between host and nematode community is not valid because of the impossibility to infer a causal effect of current nematode load on past growth and current size. Furthermore, the richness and total abundance of shedded adult Oxyurids seems to me a poor indicator of the host-nematode interaction type because it does not take into consideration the potentially differing roles of the different Oxyurid species.

Given these uncertainties, I dare to propose that the paper would actually be stronger if everything related to aim 2, L149- (and interpretation in L459-480, determining whether the nematode community in any one of the environment types was mutualistic, commensal or parasitic) is dropped from the paper. I agree that it is a very interesting possibility that there are different kinds of interactions at play and this warrants further study but unfortunately this retrospective speculation cannot really accomplish that goal in my opinion. Perhaps the paper could simply report the network analysis as that part seems to be sound and would likely make for a sufficiently strong stand-alone paper. I'm sorry I can't be more positive about this.



SECOND, I had also suggested including information on (and implications of) the ecology of nematodes beyond their potential roles as interaction partners with the host. The authors have kindly added a statement to L540- about the limitation of excluding oxyurid eggs, but the more general implications of using adult nematode shedding has not been recognized. At the very least, you should point out to less informed readers that the measure of adult nematodes in feces may or may not be reflective of the actual nematode communities inhabiting the host gut. This really is important for your network analysis. Shedding of adult nematodes is not very common and may signal digestive problems (perhaps including stress defecation due to handling). Normally, adult nematodes may not be detected in feces even if there are many of them in the digestive tract. It is a simple thing to acknowledge this limitation in your study e.g. in the last paragraph of the discussion (the study is still a valid first pass at the problem!) and e.g. propose molecular studies identifying all nematodes present in the feces based on DNA barcoding.

However, as you now clarified in the methods (L247) that all the nematodes you detected were Oxyurids, the situation is a little clearer. Oxyurid adults are probably easier to detect because they are located lower in the digestive tract. Therefore, you may have good representation of Oxyurids (possible other taxa may be underrepresented). However, as you refer to “nematodes” only throughout the paper, this fact may be lost on many readers. I suggest you clarify and reiterate in the results that only Oxyurid adults were found and point out that e.g. the network analyses all refer to Oxyurid nematodes only. ALso, calling them "endosymbionts" in the title is not founded, given that there are these multiple possible ways of interacting, better just refer to "Oxyurid nematodes" for clarity.

---

## Round 0.3 · Minor Revisions

Many thanks for this diligent revision! Both the reviewer and myself support acceptance in its current state. I would like to give you the opportunity to incorporare the last (very) minor suggestions and send in an updated version. I will then formally accept, which will then initiate the production process.

Reviewer 1 ·

Basic reporting

no comment

Experimental design

no comment

Validity of the findings

no comment

Additional comments

I thank the authors for the additional edits and patient clarifications. I hope that these have also made the paper accessible to a broader audience. I have not further concerns about the paper and I believe that in its current state it will make a valuable contribution to the field.

I have a few very minor final suggestions:

A minor English language issue: The wording changes to “association” have led to several instances where the prepositions are a bit strange. Instead of “association on” this should usually be “association with”.

L121-138: Consider restructuring this paragraph. E.g. I suggest moving the sentence in L128-130 (introducing the nematode taxa) to the beginning of the paragraph (currently first mention of ascarids seems abrupt) and rewording something in the final 2 sentences; the statements do not seem entirely logical to me.

L375: Instead of frequency, should this be number/count/diversity/richness?

L440-443: I suggest reconsidering this is long and complex sentence, perhaps split in two.

L502 -> Another/additional (possibly more parsimonious?) explanation might be stronger selective disappearance in degraded habitats, i.e. animals that were not able to grow fast enough had lower survival, and perhaps never made it to adulthood.

---

## Round 0.4 · accepted · Accept

Many thanks for considering the final reviewer comments, and for your patience and diligence in attending to the input from the reviewers and myself during the review process.